# The P2X7 receptor forms a dye-permeable pore independent of its intracellular domain but dependent on membrane lipid composition

Akira Karasawa, Kevin Michalski, Polina Mikhelzon, Toshimitsu Kawate*

Department of Molecular Medicine, Cornell University, Ithaca, United States

**Abstract** The P2X7 receptor mediates extracellular ATP signaling implicated in the development of devastating diseases such as chronic pain and cancer. Activation of the P2X7 receptor leads to opening of the characteristic dye-permeable membrane pore for molecules up to ~900 Da. However, it remains controversial what constitutes this peculiar pore and how it opens. Here we show that the panda receptor, when purified and reconstituted into liposomes, forms an intrinsic dye-permeable pore in the absence of other cellular components. Unexpectedly, we found that this pore opens independent of its unique C-terminal domain. We also found that P2X7 channel activity is facilitated by phosphatidylglycerol and sphingomyelin, but dominantly inhibited by cholesterol through direct interactions with the transmembrane domain. In combination with cell-based functional studies, our data suggest that the P2X7 receptor itself constitutes a lipid-composition dependent dye-permeable pore, whose opening is facilitated by palmitoylated cysteines near the pore-lining helix.
DOI: https://doi.org/10.7554/eLife.31186.001

**\*For correspondence:**
tk499@cornell.edu

**Competing interests:** The authors declare that no competing interests exist.

## Introduction

The P2X7 receptor is a trimeric, extracellular ATP-gated membrane channel expressed in many different cell types including immune and glial cells (*Collo et al., 1997*; *Sluyter, 2017*). Activation of the P2X7 receptor, like other P2X receptor subtypes (P2X1-6), allows small cations to traverse the plasma membrane, which triggers various intracellular signaling events such as gene expression (*North, 2002*; *Habermacher et al., 2016*; *Di Virgilio et al., 2017*). On the other hand, the P2X7 receptor exhibits several unique characteristics. In particular, its activation leads to the formation of a membrane pore that allows molecules up to ~900 Da to permeate upon ATP binding (*Steinberg et al., 1987*; *Di Virgilio, 1995*; *Surprenant et al., 1996*; *Alves et al., 2014*). This unique characteristic is supported by a number of experiments from multiple labs where heterologously overexpressed P2X7 receptors in various cell-lines take up organic cations such as ethidium and YO-PRO-1 (*Surprenant et al., 1996*; *Chessell et al., 1997*; *Rassendren et al., 1997*). Moreover, macrophages derived from a P2X7 knockout mouse fail to take up dyes after ATP application (*Solle et al., 2001*), highlighting that the P2X7 receptor is an essential component of the dye-permeable pore. This P2X7 specific pore has been implicated in multiple physiological and pathological events, including proliferation of microglia (*Monif et al., 2009*), colitic loss of enteric neurons (*Gulbransen et al., 2012*), and development of chronic pain (*Sorge et al., 2012*). However, the underlying mechanisms of the pore formation remain poorly understood.

There are two prevailing hypotheses about what actually constitutes the dye-permeable pore. The first one postulates that an ATP-release pannexin1 (Panx1) channel constitutes the pore that opens in response to P2X7 activation. This hypothesis is supported by the observations that (1) P2X7

alone is insufficient to form a dye-permeable pore (*Petrou et al., 1997*; *Klapperstück et al., 2000*), (2) P2X7 receptor expressing cells open two distinct pores for small and large ions (*Nuttle and Dubyak, 1994*; *Virginio et al., 1997*; *Faria et al., 2005*; *Jiang et al., 2005*), (3) heterologously expressed P2X7 receptors in the absence of Panx1 fail to open a dye-permeable pore (*Pelegrin and Surprenant, 2006*), (4) Panx1 inhibitors, but not P2X7 antagonists, block dye uptake (*Locovei et al., 2007*), and (5) P2X7 and Panx1 are co-immunoprecipitated in cell lysates derived from multiple cells (*Silverman et al., 2009*; *Li et al., 2011*; *Poornima et al., 2012*). However, this hypothesis has been challenged by other studies demonstrating that (1) macrophages from a Panx1 knockout mouse normally take up YO-PRO-1 (*Qu et al., 2011*), (2) Panx1 knockdown or Panx1 antagonists do not affect dye uptake (*Xu et al., 2012*; *Alberto et al., 2013*), and (3) ATP actually antagonizes Panx1 activity (*Qiu and Dahl, 2009*). In contrast to this P2X7-Panx1 complex theory, the second hypothesis postulates that the P2X7 transmembrane domain itself constitutes the large pore, which is established by dilating the small-ion permeable pore upon prolonged or repeated application of ATP. Direct permeation of dyes through P2X7 is strongly supported by the tight correlation between the charge and accessibility of the pore-lining residues and permeability of the permeant dyes (*Browne et al., 2013*). Dilation of the P2X7 membrane pore is corroborated based on the observation that a larger cation, N-methyl-D-glucamine (NMDG; Mw:195 Da), becomes more permeable over time during ATP application in whole-cell patch clamp experiments (*Virginio et al., 1999*; *Yan et al., 2008*). This P2X7 pore dilation theory, however, has also been challenged by other studies demonstrating that time-dependent NMDG permeability is most likely due to a consequence of unintended ion accumulation in whole-cell patch clamp experiments (*Li et al., 2015*). Furthermore, studies using single channel recordings provided compelling evidence that the unitary conductance of the P2X7 receptor remains constant over time (*Riedel et al., 2007*; *Harkat et al., 2017*; *Pippel et al., 2017*). These studies disagree with the pore-dilation model and suggest that the P2X7 receptor immediately opens a pore that is large enough for NMDG to permeate.

Setting aside the controversial building block of the P2X7 triggered dye-permeable pore, it has been consistently reported that dye-uptake strictly depends on its unique C-terminal domain (CTD), which exists only in the P2X7 subtype (~250 AA). Indeed, cells expressing C-terminally truncated versions of the P2X7 receptor fail to take up large molecules, while small cations (e.g. Na$^+$ and Ca$^{2+}$) remain permeable (*Surprenant et al., 1996*; *Rassendren et al., 1997*; *Cheewatrakoolpong et al., 2005*; *Adinolfi et al., 2010*; *Cervetto et al., 2013*). While the primary sequence of the CTD is dissimilar to those of any known proteins, it harbors a number of potential binding motifs for molecules such as β-arrestin, bacterial lipopolysaccharide, calmodulin, and kinases (*Denlinger et al., 2001*; *Kim et al., 2001*; *Roger et al., 2008*; *Roger et al., 2010*; *Costa-Junior et al., 2011*). It has also been suggested to be involved in proper assembly and trafficking (*Denlinger et al., 2003*; *Smart et al., 2003*; *Gonnord et al., 2009*; *Bradley et al., 2010*). Interestingly, there is a stretch of 18 amino acid residues rich in cysteine (six cysteines) located at the beginning of the CTD (*Figure 1A*), right after the second transmembrane helix. This Cys-rich region (CRR) has been proposed to play major roles in lipid raft association (*Gonnord et al., 2009*), cholesterol binding (*Robinson et al., 2014*; *Murrell-Lagnado, 2017*), and channel gating (*Allsopp and Evans, 2015*). However, it remains unclear why the CTD is required for the formation of the P2X7 dependent pore.

In this study, we investigated the P2X7 dependent dye uptake using proteoliposomes reconstituted with purified P2X7 receptors. Our *in vitro* system is advantageous in that it allows all components of the liposome, such as lipid composition and ratios, to be freely controlled. Using this system, we discovered that P2X7 alone is sufficient to form a dye-permeable pore, and we showed that the activity of P2X7 depends heavily on the lipid composition. Surprisingly, both the CTD and the N-terminal domain (NTD) seemed indispensable for dye-uptake itself in certain lipid compositions. In combination with cellular dye-uptake assays, we propose the CRR in the CTD facilitates P2X7 receptor activation.

## Results

### P2X7-ΔNC alone is sufficient for dye-uptake in proteoliposomes

The uncertain nature of the P2X7 dependent pore may be attributed to uncharacterized cellular components that exist inconsistently in different cell types. To unambiguously determine whether

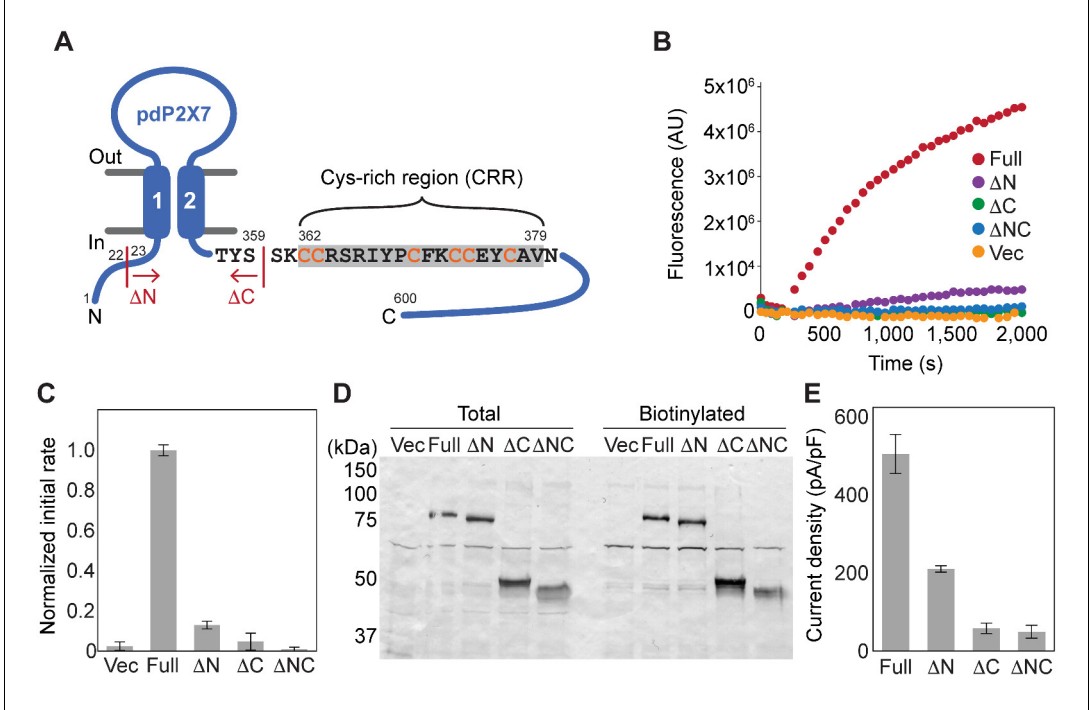

**Figure 1.** Characterization of deletion constructs of pdP2X7. (**A**) Schematic representation of pdP2X7. The start or end position of each deletion construct is shown. ΔN: Δ1–22; ΔC: Δ360–600; ΔNC: Δ1–22/Δ360–600. (**B and C**) ATP-evoked YO-PRO-1 uptake in HEK293 cells expressing each construct. Representative results are shown in (**B**). The initial rate of YO-PRO-1 uptake after ATP application was calculated and normalized to the initial rate of the full-length pdP2X7 (**C**). The bars represent the means of at least five independent experiments and the error bars represent SEM. Vec; vector transfected HEK cells. (**D**) Surface expression of P2X7 constructs. FLAG tag was attached to the N-terminus of P2X7 constructs and expressed in HEK293 cells. Surface expressed protein was biotinylated and probed by western blotting. Total protein sample after solubilization (total) and eluted sample from resin (biotinylated) are shown. (**E**) Current densities of the pdP2X7 constructs. Currents were obtained by whole cell patch clamp recordings triggered by 1 mM ATP. Membrane potential was held at −60 mV. Bars represent the means of at least four measurements and the error bars represent SEM.

DOI: https://doi.org/10.7554/eLife.31186.002

The following figure supplement is available for figure 1:

**Figure supplement 1.** Whole cell patch clamp recordings of pdP2X7 deletion constructs.

DOI: https://doi.org/10.7554/eLife.31186.003

the P2X7 receptor is the conduit for large molecules, we set out to establish an *in vitro* dye-uptake system using proteoliposomes reconstituted with purified P2X7 receptors. We took advantage of the panda P2X7 (pdP2X7) that is ~85% identical to the human P2X7 (hP2X7), mediates YO-PRO-1 (Mw: 376 Da without iodide) uptake in HEK293 cells, and has been successfully purified for crystallization (***Karasawa and Kawate, 2016***). When expressed in HEK293 cells, truncated versions of pdP2X7 at the N-terminus (ΔN; Δ1–22) or at the C-terminus (ΔC; Δ360–600) diminished the YO-PRO-1 uptake by more than 10 fold compared to the full-length receptor (***Figure 1B and C***). When both termini were truncated (ΔNC), there was no detectable YO-PRO-1 uptake (***Figure 1B and C***). Surface biotinylation experiments showed that the expression levels of truncated receptors were comparable or even slightly higher for ΔC, suggesting that the diminished ATP-triggered YO-PRO-1 uptake was due to reduced channel activity but not due to lower surface expression (***Figure 1D***). Current densities from whole-cell patch clamp recordings revealed a similar pattern, though the degree of current reduction was only ~50% for ΔN, and both ΔC and ΔNC gave rise to currents that were less than ~10% of the full-length channel (***Figure 1E*** and ***Figure 1—figure supplement 1***). These results suggest that the CTD and the NTD (though to a lesser extent) play important roles in both dye-uptake and small ion permeation in HEK293 cells.

We subsequently investigated the channel activity of pdP2X7 in proteoliposomes. To assess whether either terminus of pdP2X7 is required for the channel activity *in vitro*, we purified and

reconstituted pdP2X7 protein that is truncated at both termini (pdP2X7-ΔNC) into liposomes composed of two commonly used synthetic lipids for ion channel reconstitution (1-palmitoyl-2-oleoyl-sn-glycero-3-phosphoethanolamine (POPE) and 1-palmitoyl-2-oleoyl-sn-glycero-3-phospho-(1'-rac-glycerol) (POPG)). Orientation of the reconstituted pdP2X7-ΔNC was mostly outside-out, as externally applied recombinant EndoH removed the extracellular carbohydrates attached through posttranslational glycosylation (*Figure 2—figure supplement 1*). We first examined $Ca^{2+}$ permeability using liposomes containing a $Ca^{2+}$-sensing fluorescent molecule, Fluo-4 (*Figure 2A*). When pdP2X7-ΔNC containing liposomes were treated with ATP, the $Ca^{2+}$ signal increased in a dose-dependent manner (*Figure 2B and C*). The apparent low channel activity of pdP2X7-ΔNC in these experiments is likely due to the diminished concentrations of free ATP ($ATP^{4-}$) resulting from the formation of calcium-ATP complexes. Indeed, $EC_{50}$ of the calculated free ATP was 29 µM, which was comparable with that obtained for the crystallized version of pdP2X7 (pdP2X7-cryst; $EC_{50}$: 40 µM) in whole-cell recordings without calcium in the extracellular solution (*Karasawa and Kawate, 2016*). This

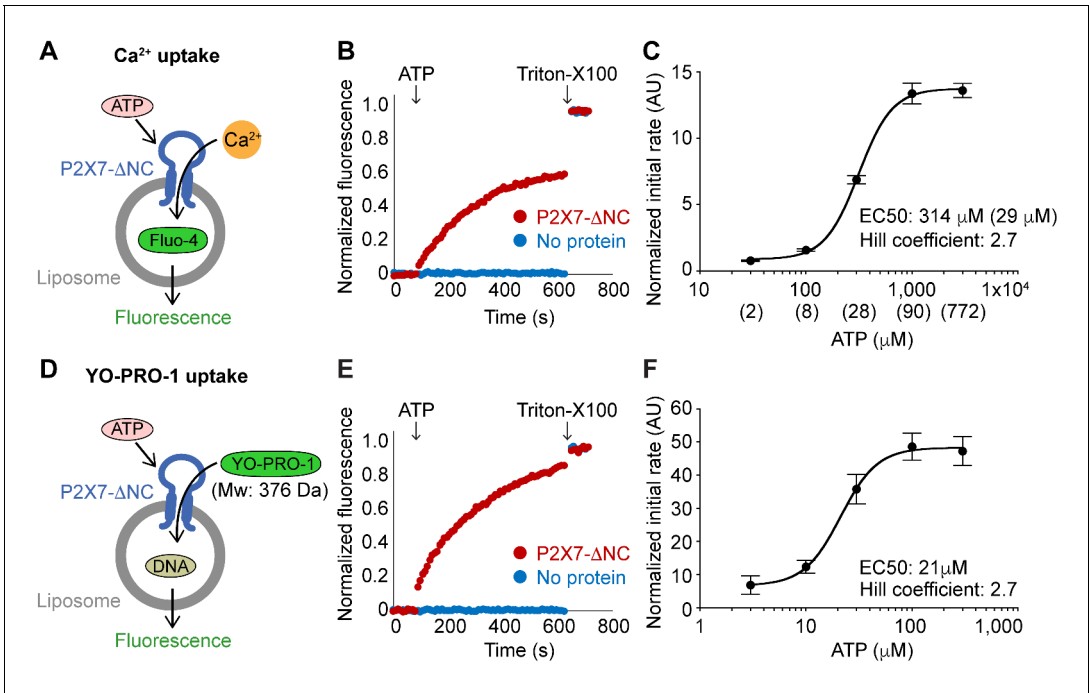

**Figure 2.** Reconstituted pdP2X7-ΔNC takes up both $Ca^{2+}$ and YO-PRO-1 in liposomes. (A) Schematic representation of $Ca^{2+}$ uptake assay. Purified pdP2X7-ΔNC was reconstituted into liposomes and the $Ca^{2+}$ indicator Fluo-4 was incorporated by freeze-thaw cycles. $Ca^{2+}$ influx was monitored after ATP application by measuring Fluo-4 fluorescence. (B) pdP2X7-ΔNC mediated $Ca^{2+}$ uptake triggered by ATP. pdP2X7-ΔNC was reconstituted into POPE:POPG 3:1 (w/w) liposomes at 1:100 ratio of protein to liposome and Fluo-4 fluorescence was monitored before and after application of 1 mM ATP (red). Fluorescence was normalized to the total fluorescence obtained after application of 1% Triton-X100. Empty liposomes (no protein) with the same composition are shown as a control (blue). (C) ATP dose-responses of pdP2X7-ΔNC in liposome for $Ca^{2+}$ uptake. Plots were made using the means of six independent experiments and the error bars represent SEM. Dose-response curves were fit to the Hill equation. The numbers in parentheses are calculated free ATP ($ATP^{4-}$) concentrations using Max Chelator (http://maxchelator. stanford.edu/). (D) Schematic representation of YO-PRO-1 uptake assay. Purified pdP2X7-ΔNC was reconstituted into liposomes and a 20-mer double-strand DNA was incorporated by freeze-thaw cycles. YO-PRO-1 influx was measured by an increase of YO-PRO-1 fluorescence triggered by DNA-binding. (E) pdP2X7-ΔNC mediated YO-PRO-1 uptake triggered by 1 mM ATP. pdP2X7-ΔNC was reconstituted into POPE:POPG 3:1 (w/w) liposomes and YO-PRO-1 fluorescence was monitored before and after application of 1 mM ATP (red). Fluorescence was normalized to the total fluorescence obtained after applying 1% Triton-X100. Empty liposomes (no protein) with the same composition are shown as a control (blue). (F) ATP dose-response of pdP2X7-ΔNC in liposomes determined by YO-PRO-1 uptake. The plots were made using the means of four independent experiments and the error bars represent SEM. Dose response curves were fit to the Hill equation.

DOI: https://doi.org/10.7554/eLife.31186.004

The following figure supplement is available for figure 2:

**Figure supplement 1.** Outside-out configuration of pdP2X7-ΔNC in liposomes.
DOI: https://doi.org/10.7554/eLife.31186.005

suggests that the P2X7 receptor, like the P2X2 and 4 subtypes, requires free ATP for its activation (*Klapperstück et al., 2001*; *Li et al., 2013*). The ATP dose-dependent $Ca^{2+}$-uptake indicates that the reconstituted pdP2X7-ΔNC is functional and both the NTD and the CTD are not required for permeation by small cations.

We next assessed whether pdP2X7-ΔNC can mediate uptake of a larger cation in the absence of other cellular components. Like in a cell-based system, YO-PRO-1 uptake was followed by monitoring the change in fluorescence upon binding to DNA that was encapsulated into liposomes (*Figure 2D*). To our surprise, the liposomes reconstituted with pdP2X7-ΔNC presented a robust YO-PRO-1 uptake in a dose dependent manner (*Figure 2E and F*). There was no time-lag in YO-PRO-1 uptake after ATP application and the response was monophasic. The Hill coefficient was 2.7 and the $EC_{50}$ was 21 µM, which were comparable with the $Ca^{2+}$ uptake assay (*Figure 2B and C*). These results demonstrate that pdP2X7 alone is sufficient to form a membrane pore large enough for YO-PRO-1 to permeate and that neither N- or C-terminus is required for this process. Furthermore, the immediate and monophasic increase of the fluorescence signal indicate that pore dilation is unlikely.

## P2X7-ΔNC activity heavily depends on lipid compositions

Why does pdP2X7-ΔNC not form a dye-permeable pore in HEK293 cells? Because only a few, if any, intracellular residues exist in this truncated construct, it is unlikely that other intracellular proteins down regulate the function of pdP2X7-ΔNC. We speculated that HEK293 membranes may contain molecules that inhibit the P2X7 receptor through its transmembrane domain. In particular, we suspected that a specific kind of lipid present in the plasma membrane may antagonize pore formation. To test this idea, we reconstituted pdP2X7-ΔNC into liposomes that contain the major lipids constituting the plasma membrane of HEK293 cells (termed here as 'HEK lipids'; 28.5% 1-palmitoyl-2-oleoyl-sn-glycero-3-phosphocholine (POPC), 16.5% POPE, 18% 1-palmitoyl-2-oleoyl-sn-glycero-3-phospho-L-serine (POPS), 2.25% POPG, 9.75% Sphingomyelin (SM), and 25% cholesterol). In those lipids, uptake of $Ca^{2+}$ or YO-PRO-1 by pdP2X7-ΔNC was drastically attenuated (*Figure 3A and B*), which was in contrast to the robust channel activity in lipids containing only POPE and POPG (*Figure 3C*). The reconstitution efficiency was comparable to that of POPE/POPG liposomes (*Figure 2—figure supplement 1*). These results support the idea that a component of the HEK lipid mixture antagonizes P2X7 function.

To identify which lipid(s) antagonizes pdP2X7-ΔNC, we next measured YO-PRO-1 uptake in liposomes enriched with each component of the HEK lipids. Because liposomes containing only one of those lipids were unstable and irregular in size, we included 25% POPC and 25% POPE in each liposome. SDS-PAGE confirmed that the reconstitution efficiency was comparable to each other (*Figure 3—figure supplement 1*). Remarkably, only weak YO-PRO-1 uptake activity was observed with the control liposomes composed of 50% POPC and 50% POPE (*Figure 3D*). Likewise, we observed similarly low levels of YO-PRO-1 uptake activities with the liposomes containing 50% POPC, POPE, or cholesterol (*Figure 3D*). Conversely, when the liposomes contained 50% POPG or SM, YO-PRO-1 uptake was substantially enhanced. POPG especially boosted the YO-PRO-1 uptake by ~7 fold compared to the control. These results suggest that POPG and SM facilitate the activity of pdP2X7-ΔNC and that POPE, POPC, and cholesterol counteract the facilitating action of those lipids. However, POPE is unlikely a dominant antagonist of P2X7, as strong dye-uptake was observed with pdP2X7-Δ NC reconstituted in 75% POPE and 25% POPG (*Figures 2* and *3D*). POPC is also unlikely to be a dominant antagonist as the liposomes exhibiting higher levels of YO-PRO-1 uptake (i.e. POPE or SM) also contained 25% POPC. Cholesterol therefore seems to be the dominant antagonist of pdP2X7-ΔNC in the HEK lipids. This is consistent with weak dye-uptake in HEK lipids that contain 25% cholesterol, despite also containing a total of 30% of the P2X7 facilitating lipids (i.e. POPG and SM).

To confirm the dominant and inhibitory role of cholesterol in HEK lipids, we measured YO-PRO-1 uptake from pdP2X7-ΔNC that was reconstituted into HEK mimetic liposomes containing different amounts of cholesterol. The liposomes without cholesterol showed as strong YO-PRO-1 uptake as the POPE/POPG liposomes (*Figure 4A and B*), consistent with the idea that POPG and SM facilitate P2X7 channel activity. As we expected, we observed less dye-uptake as the concentration of cholesterol increased until it reached 15% where the inhibitory effect saturated (*Figure 4A and B*). The same pattern was observed for $Ca^{2+}$ uptake (*Figure 4—figure supplement 1*). These results indicate that the inhibitory action of cholesterol can be seen at a concentration as low as 5% and that the

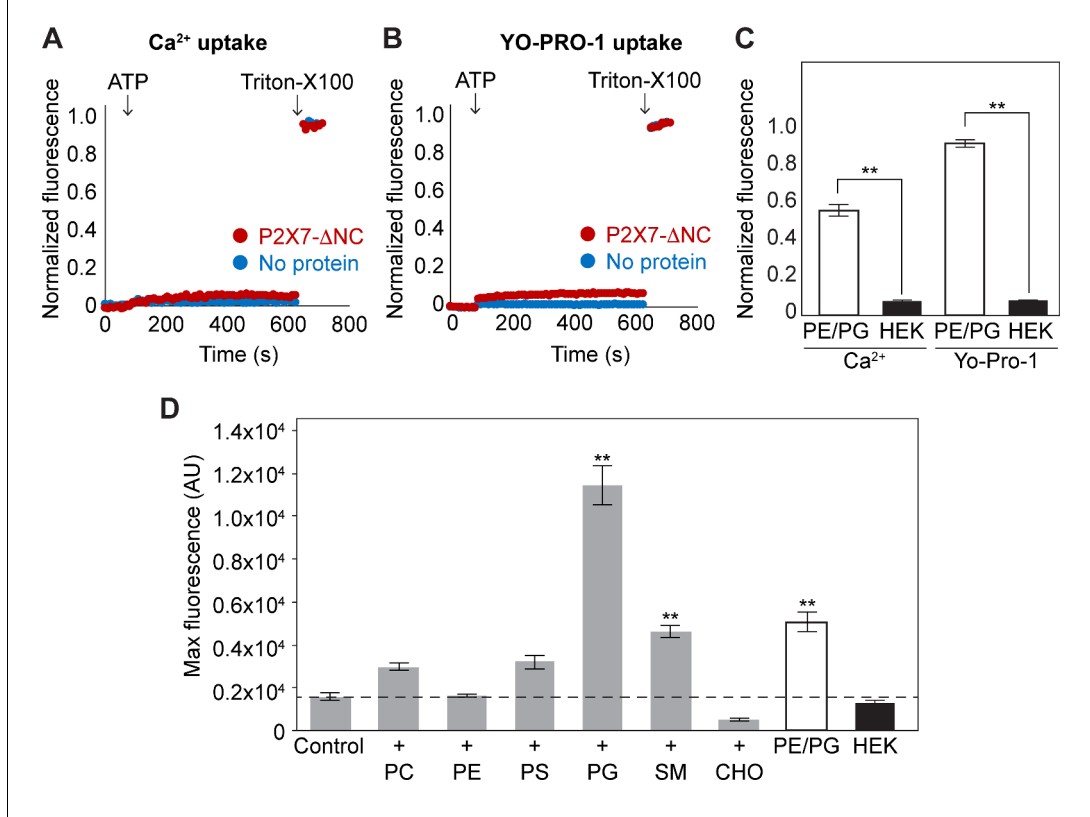

**Figure 3.** Lipid-dependent YO-PRO-1 uptake of pdP2X7-ΔNC in liposomes. (A) and (B) $Ca^{2+}$ and YO-PRO-1 uptake of pdP2X7-ΔNC in HEK293-mimetic liposomes. pdP2X7-ΔNC was reconstituted into liposomes composed of 28.5% POPC, 16.5% POPE, 2.25% POPG,18% POPS, 9.75% sphingomyelin, and 25% cholesterol. Fluo-4 (A) or YO-PRO-1 (B) fluorescence was monitored before and after application of 1 mM ATP (red). Empty liposomes (no protein) with the same composition are shown as a control (blue). Fluorescence was normalized to the total fluorescence obtained after application of 1% Triton-X100. (C) Comparison between $Ca^{2+}$ and YO-PRO-1 uptake in POPE/POPG or HEK293-mimetic liposomes. The bars represent the means of maximum fluorescence after ATP application normalized to the total fluorescence obtained after application of 1% Triton-X100. The means were obtained from at least four independent experiments and error bars represent SEM. Asterisks indicate a significant difference (p<0.01) between the PE/PG and HEK293-mimetic liposomes as determined by student's t-test. (D) Lipid-dependent YO-PRO-1 uptake of pdP2X7-ΔNC. Purified pdP2X7-ΔNC was reconstituted into liposomes with different lipid compositions. The bars represent the means of maximum fluorescence after ATP application from four independent experiments and the error bars represent SEM. Control: 50% POPE and 50% POPC;+PC: 25% POPE and 75% POPC;+PE: 75% POPE and 25% POPC;+PS: 25% POPE, 25% POPC, and 50% POPS;+PG: 25% POPE, 25% POPC, and 50% POPG;+SM: 25% POPE, 25% POPC, and 50% sphingomyelin;+CHO: 25% POPE, 25% POPC, and 50% cholesterol; PE/PG: 75% POPE and 25% POPG; HEK: 28.5% POPC, 16.5% POPE, 2.25% POPG,18% POPS, 9.75% sphingomyelin, and 25% cholesterol. Asterisks indicate significant difference compared to the control (p<0.01) as determined by one way ANOVA followed by Dunnett's test.

DOI: https://doi.org/10.7554/eLife.31186.006

The following figure supplement is available for figure 3:

**Figure supplement 1.** Comparable reconstitution efficiency of pdP2X7-ΔNC in liposomes with different lipid compositions.
DOI: https://doi.org/10.7554/eLife.31186.007

inhibitory effect saturates around 15%. Considering that the plasma membrane of HEK293 cells normally includes ~25% cholesterol (*Dawaliby et al., 2016*), it makes sense why pdP2X7-ΔNC channels hardly open in those cells, resulting in low YO-PRO-1 uptake (*Figure 1B and C*) and low current density (*Figure 1E*). Altogether, our results demonstrate that P2X7 receptor activity depends heavily on lipid compositions. Especially, concentrations of POPG, SM, and cholesterol seem to play important roles in regulating the P2X7 receptor activity.

## Cholesterol inhibits P2X7 by binding to the transmembrane domain

How does cholesterol inhibit pdP2X7-ΔNC? Though cholesterol has been suggested to interact with the intracellular juxtamembrane domains (*Robinson et al., 2014*), other mechanisms must also exist

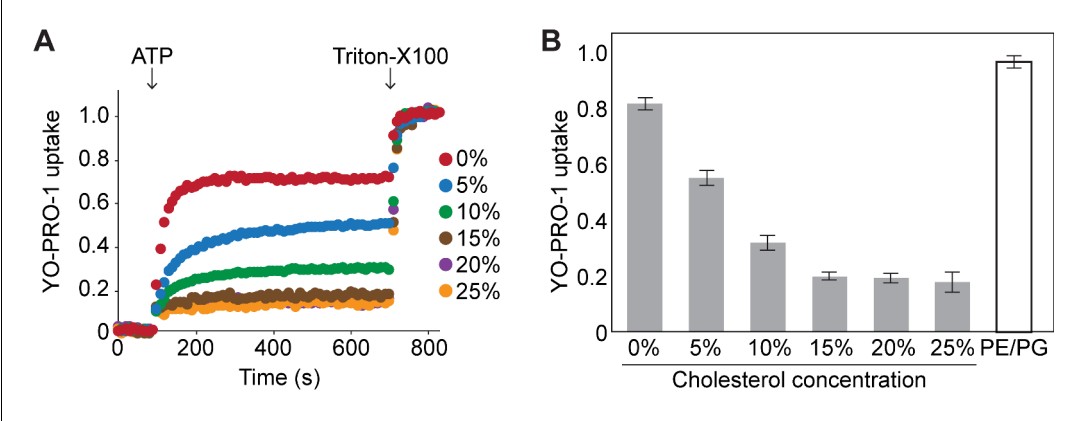

**Figure 4.** Cholesterol inhibits YO-PRO-1 uptake of pdP2X7-ΔNC in a dose dependent manner. (**A**) Purified pdP2X7-ΔNC was reconstituted into liposomes with different cholesterol compositions and ATP-evoked YO-PRO-1 uptake was measured. Liposomes were prepared by adding different amounts of cholesterol (final 5–25%) to the lipid mixture composed of 38% POPC, 22% POPE, 3% POPG, 24% POPS, and 13% sphingomyelin. (**B**) Comparison of YO-PRO-1 uptake of proteoliposomes composed of different cholesterol contents. Bars represent the mean maximum fluorescence after ATP application from nine independent experiments and the error bars represent SEM. Fluorescence was normalized to the total fluorescence obtained after application of 1% Triton-X100. PE/PG: liposome containing 75% POPE and 25% POPG.

DOI: https://doi.org/10.7554/eLife.31186.008

The following figure supplement is available for figure 4:

**Figure supplement 1.** Cholesterol also inhibits $Ca^{2+}$ uptake of pdP2X7-ΔNC in a dose dependent manner.

DOI: https://doi.org/10.7554/eLife.31186.009

as the intracellular domains are absent in pdP2X7-ΔNC. Because cholesterol is a well-known lipid that structures ordered micro-domains and increases membrane rigidity (*Hancock, 2006*; *Ikonen, 2008*; *de Meyer and Smit, 2009*), we first assessed whether cholesterol antagonizes pdP2X7-ΔNC function by increasing the membrane rigidity. To quantify the membrane rigidity in our proteoliposome-based system, we took advantage of 1,6-diphenyl-1,3,5-hexatriene (DPH; *Figure 5A*), which gives rise to fluorescence anisotropy as membrane rigidity increases (*Dawaliby et al., 2016*). As shown in *Figure 5B*, DPH anisotropy increased as the concentration of cholesterol increased from 0% to 25% in liposomes composed of HEK lipids. This suggests that cholesterol indeed enhances membrane rigidity in a dose dependent manner. In contrast, no significant fluorescence anisotropy was observed when DPH was incubated with the liposomes enriched with POPC, POPE, POPS, or POPG (*Figure 5C*). Notably, the SM-enriched liposome increased the level of DPH anisotropy, indicating that SM also enhances the membrane rigidity. Because SM-enriched liposomes facilitate the channel activity of pdP2X7-ΔNC (*Figure 3D*), this result suggests that membrane rigidity per se unlikely regulates the P2X7 receptor activity. Supporting this, linear regression analysis revealed no significant correlation ($R^2 = 0.044$) between DPH anisotropy and YO-PRO-1 uptake activity in the absence of cholesterol (*Figure 5D*). These results indicate that the inhibitory effect of cholesterol derives from an alternative mechanism and that the increased membrane rigidity probably plays a lesser role, if any.

We next tested whether cholesterol directly interacts with pdP2X7-ΔNC. To quantify cholesterol binding *in vitro*, we used a fluorescent analogue, 22-NBD-cholesterol (*Figure 6A*), which is commonly used for studying cholesterol binding proteins such as NPC1 (*Liu et al., 2009*). The fluorescence emission spectrum showed substantial increase in fluorescence when 22-NBD-cholesterol was incubated with pdP2X7-ΔNC (*Figure 6B*). The no protein control showed much weaker fluorescence, suggesting that the enhanced fluorescence was due to the 22-NBD-cholesterol binding to pdP2X7-ΔNC but not to detergent micelles. Importantly, binding of 22-NBD-cholesterol to pdP2X7-ΔNC was dose-dependent, supporting specific interactions between cholesterol and pdP2X7-ΔNC (*Figure 6C*). Together, these results suggest that cholesterol inhibits pdP2X7-ΔNC by directly interacting with its transmembrane domain, rather than by indirectly changing the membrane fluidity.

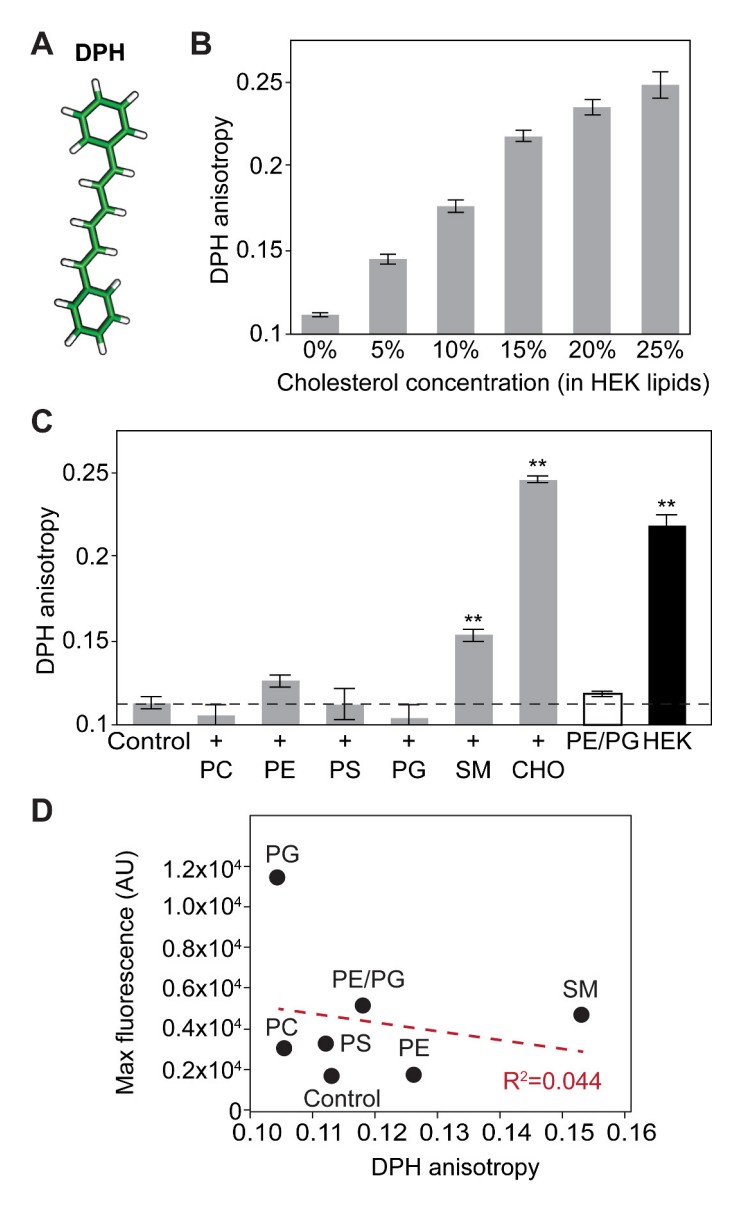

**Figure 5.** Membrane rigidity of liposomes composed of different lipids monitored by DPH fluorescence anisotropy. (**A**) Stick representation of 1,6-diphenyl-1,3,5-hexatriene (DPH). (**B**) DPH anisotropy from pdP2X7-ΔNC reconstituted liposomes composed of HEK lipids with designated amount of cholesterol. DPH anisotropy was obtained at Ex: 358 nm and Em: 429 nm. (**C**) DPH anisotropy from pdP2X7-ΔNC reconstituted liposomes composed of different kinds of lipids as described in *Figure 3*. Asterisks indicate significant difference compared to wildtype or the no protein control (p<0.01) as determined by one-way ANOVA followed by Dunnett's test. (**D**) Linear regression analysis of maximum fluorescence and DPH anisotropy. The shown values are the averages of 4–5 independent experiments.

DOI: https://doi.org/10.7554/eLife.31186.010

## The C-terminal Cys-rich region counteracts the inhibitory effect of cholesterol

We have demonstrated that a high concentration (~25%) of cholesterol in the HEK293 cell membrane almost completely abolishes the channel activity of pdP2X7-ΔNC. But the full-length pdP2X7 receptor maintains its channel activity in HKE293 cells (*Figure 1B,C and E*). How does the full-length pdP2X7 escape from the inhibitory effect of cholesterol? Previous studies demonstrated

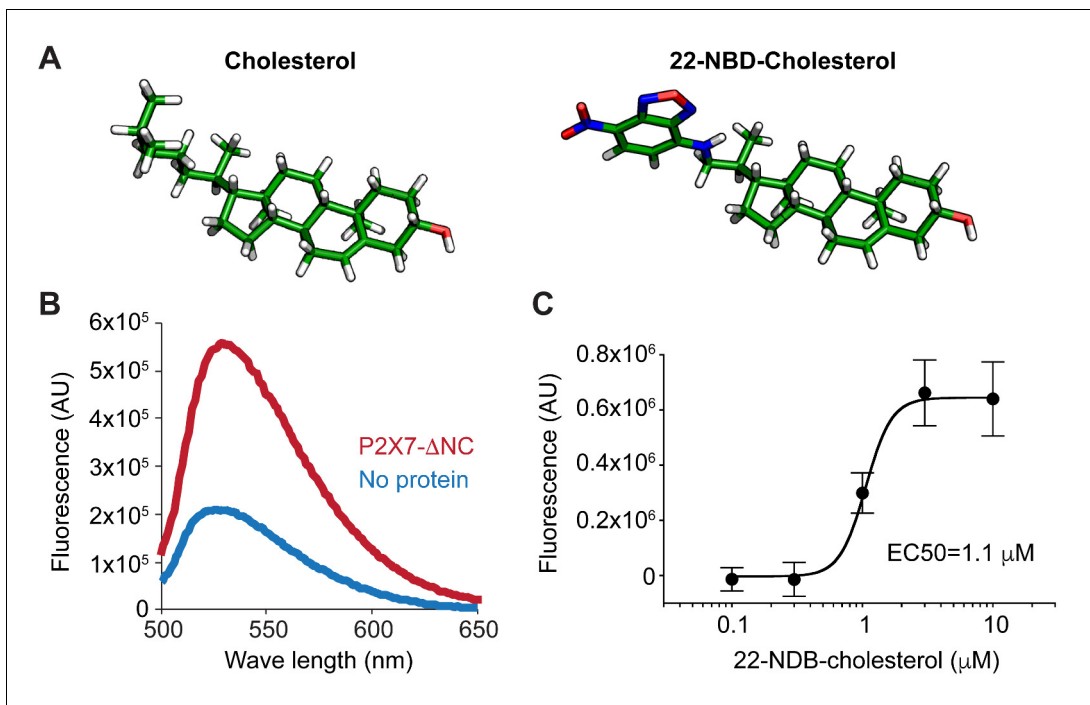

**Figure 6.** Cholesterol binds to purified pdP2X7-ΔNC. (**A**) Stick representation of cholesterol and 22-NBD-cholesterol. (**B**) Fluorescence spectra of 22-NBD-cholesterol after incubating with pdP2X7-ΔNC (red) or without protein (blue) for 1 hr at room temperature. (**C**) Dose-dependent 22-NBD-cholesterol binding to pdP2X7-ΔNC. The plots were made using the means of five independent experiments and the error bars represent SEM. Dose response curves were fit with the Hill equation.

DOI: https://doi.org/10.7554/eLife.31186.011

that a deletion of the 18 amino-acid Cys-rich region (CRR) in hP2X7 diminishes its channel activity without affecting the surface expression, suggesting an important role of the CRR in P2X7 activity (**Robinson et al., 2014**; **Allsopp and Evans, 2015**). However, another study proposed otherwise; the CRR-deficient rat P2X7 receptor maintains full channel activity (**Jiang et al., 2005**). To further explore the potential role of the CRR in facilitating the P2X7 receptor activity, we extended the pdP2X7-ΔNC by 18 amino acids to include the CRR (pd) and measured YO-PRO-1 uptake in HEK293 cells. Remarkably, pd showed small but significant dye-uptake activity, supporting that the CRR facilitates P2X7 channel activity (**Figure 7A**). When the level of cholesterol was reduced by treating HEK293 cells with methyl-β-cyclodextrin (MCD), YO-PRO-1 uptake dramatically increased with the constructs that include the CRR (i.e. full-length, ΔN, and ΔNC+CRR). In particular, dye-uptake activity of pd was recovered to the level comparable to the full-length pdP2X7, substantiating that the CRR is a crucial region of the C-terminus for counteracting the inhibitory action of cholesterol (**Figure 7A**). The effect of MCD was cholesterol dependent, as no significant enhancement in dye-uptake was observed when pdP2X7-ΔNC+CRR was expressed in insect Sf9 cells that lack cholesterol in the plasma membrane (**Figure 7B**). In those cells, pdP2X7-ΔNC presented strong YO-PRO-1 uptake activity, consistent with our reconstitution experiments in the absence of cholesterol (**Figures 3** and **4**).

We next set out to identify the minimal region of CRR that confers the P2X7 facilitating activity. Because YO-PRO-1 uptake of pdP2X7-ΔNC+CRR was weak in the absence of MCD, we compared the activity of truncated constructs (**Figure 8A**; CRRa to CRRe) in the presence of MCD. Whereas strong YO-PRO-1 uptake was observed for ΔNC + CRRa, b, and c, further truncated constructs (CRRd and e) showed much weaker activity (**Figure 8B**). These results indicate that the first nine residues in CRR (i.e. between C362 to C370) harbor an essential element for facilitating P2X7 receptor channel activity.

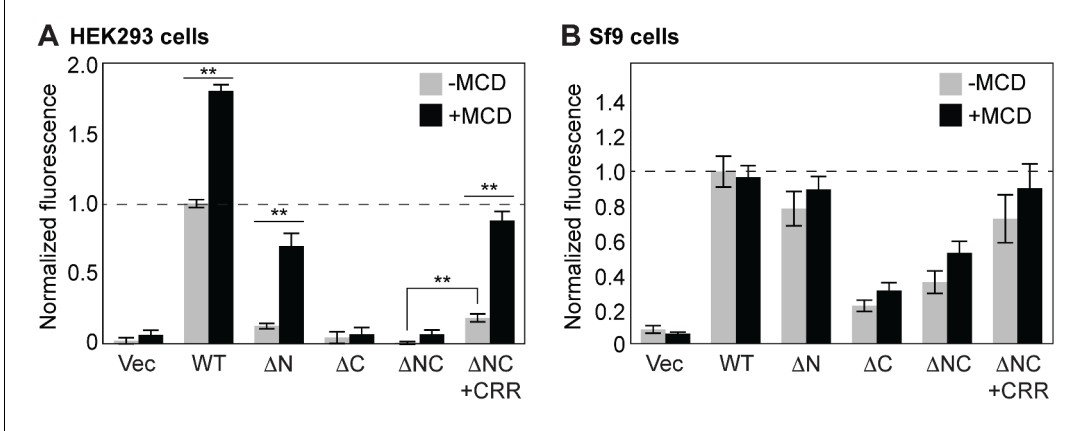

**Figure 7.** CRR facilitates YO-PRO-1 uptake in the presence of MCD. (**A**) ATP-evoked YO-PRO-1 uptake from HEK293 cells expressing each construct. HEK293 cells were treated with 5 mM Methyl-β-cyclodextran (MCD) for 1 hr at 37°C before the assay was performed. The initial rate of YO-PRO-1 uptake in the presence (black) or absence (grey) of MCD was calculated and normalized by the initial rate of the full length protein without MCD. The bars represent the means of at least five independent experiments and the error bars represent SEM. (**B**) ATP-evoked YO-PRO-1 uptake from Sf9 cells expressing each construct. Bars represent the means of the normalized initial rate from five independent experiments and the error bars represent SEM. Asterisks indicate significant difference between YO-PRO-1 uptake with and without MCD treatment (p<0.01) as determined by one way ANOVA followed by Dunnett's test. No significant fluorescence change was observed for YO-PRO-1 uptake from Sf9 cells with or without MCD.
DOI: https://doi.org/10.7554/eLife.31186.012

Which cysteine residues in the CRR, if any, play an important role in P2X7 receptor facilitation? To find out those residues, we systematically mutated each cysteine in CRR and measured YO-PRO-1 uptake in the presence of MCD. Consistent with the dye-uptake experiments using the truncation constructs, the cysteine mutants outside of the first nine amino acids in CRR (i.e. C373S, C374S, and C377S) all showed strong channel activity (*Figure 8C*). In contrast, the C362S or C363S mutants showed ~40% to 60% less activity than P2X7-ΔNC+CRR. Interestingly, the C370S mutant presented a strong channel activity despite that the truncation of the CRR before this residue (CRRd) reduced the dye-uptake by more than 50% (*Figure 8B*). When both C362S and C363S were combined, the channel activity was almost completely abolished (*Figure 8C*). These results indicate that none of the individual cysteines were essential, but C362 and C363 play dominant roles in facilitating the P2X7 receptor activity. Indeed, when these two cysteine residues were mutated to serines in the full-length pdP2X7 receptor, no YO-PRO-1 activity was observed (*Figure 8—figure supplement 1A*). This result was consistent with the C363A mutant of hP2X7, which also showed little dye-uptake (*Robinson et al., 2014*). Because the surface expression was unaffected by the C362S/C363S mutations, it is likely that these cysteines play roles in enhancing P2X7 channel opening, rather than affecting trafficking or assembly (*Figure 8—figure supplement 1B*).

Multiple cysteine residues in the CTD, including the ones in CRR, have been suggested to be posttranslationally modified by palmitoylation (*Gonnord et al., 2009*). We wondered whether the P2X7 facilitating function of CRR is mediated by palmitoylated cysteines in this region. To test this idea, we analyzed the level of palmitoylation on pdP2X7-ΔNC+CRRc that includes the minimal length of the CRR necessary for facilitating the channel activity in the presence of MCD (i.e. the first nine residues in CRR including C362/C363/C370). We metabolically labeled pdP2X7-ΔNC+CRRc with a palmitic-acid analogue, 17-octadecynoic acid (17-ODYA), pulled down using FLAG affinity chromatography, and specifically labeled the attached 17-ODYA with Alexa 488 using click chemistry. This technique has been widely used for detecting palmitoylation of many proteins including the RalA G-protein (*Martin, 2013*; *Nishimura and Linder, 2013*). We found that pdP2X7-ΔNC+CRRc is heavily modified with 17-ODYA, which resulted in a much stronger fluorescence than RalA (*Figure 9*). Because the pdP2X7-ΔNC, which is only nine amino acids shorter than pdP2X7-ΔNC+CRRc was not modified by 17-ODYA, it is likely that the cysteine residues in CRRc are indeed palmitoylated (*Figure 9*). In combination with the cysteine mutagenesis study, these results suggest that palmitoylation of at least one of the C362, C363, C370 plays an important role in enhancing the

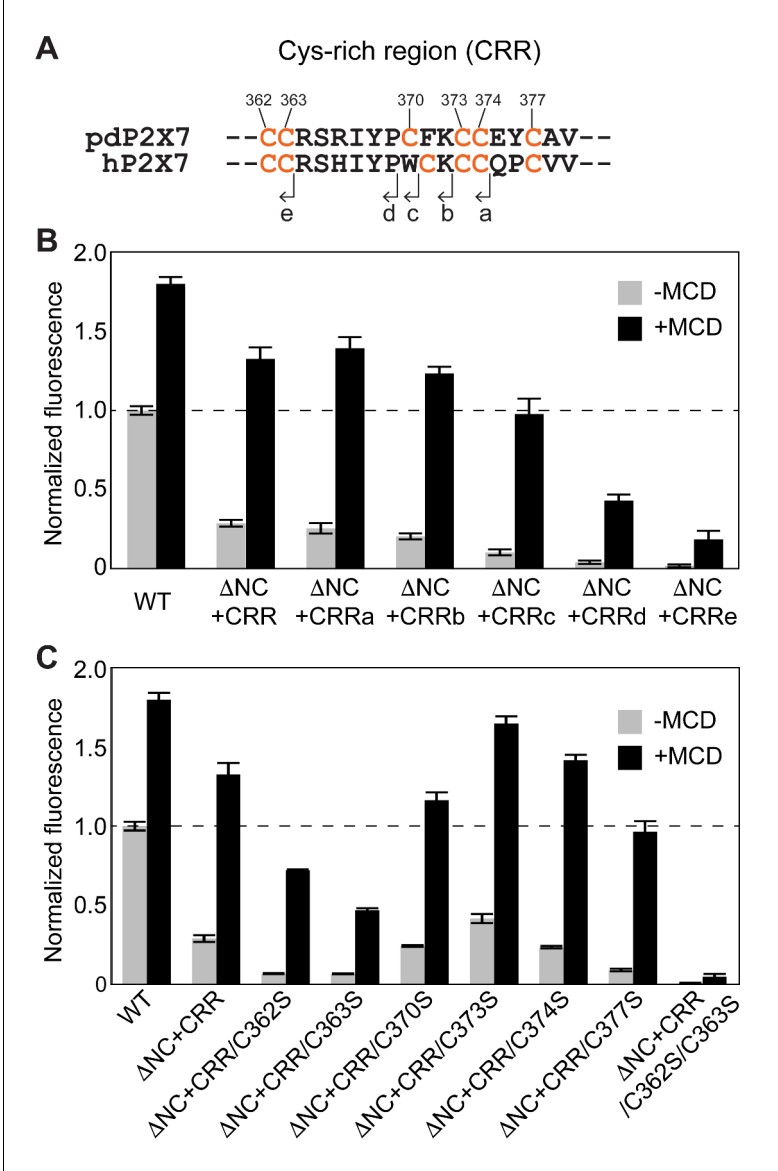

**Figure 8.** Critical region of CRR for facilitating P2X7 mediated YO-PRO-1 uptake. (**A**) Amino acid sequences of the CRR of panda and human P2X7 receptors. The end position of each deletion construct (CRRa-e) is shown. (**B**) ATP-evoked YO-PRO-1 uptake from HEK293 cells expressing each pdP2X7 deletion construct in the presence (black) or absence (grey) of MCD. Initial rate of YO-PRO-1 uptake was calculated and normalized by the initial rate of the full length protein without MCD. The bars represent the means of at least eight independent experiments and error bars represent SEM. (**C**) ATP-evoked YO-PRO-1 uptake of HEK293 cells expressing Cys knockout mutants in the presence (black) or absence (grey) of MCD. The bars represent the means of at least four independent experiments and the error bars represent SEM.

DOI: https://doi.org/10.7554/eLife.31186.013

The following figure supplement is available for figure 8:

**Figure supplement 1.** C362S/C363S mutation abolishes YO-PRO-1 uptake of pdP2X7 in HEK293 cells.

DOI: https://doi.org/10.7554/eLife.31186.014

P2X7 channel activity. Altogether, our results demonstrated that palmitoylation of CRR in the C-terminus facilitates the channel opening of the P2X7 receptor, which contributes to counteract the inhibitory effect of cholesterol in the plasma membrane.

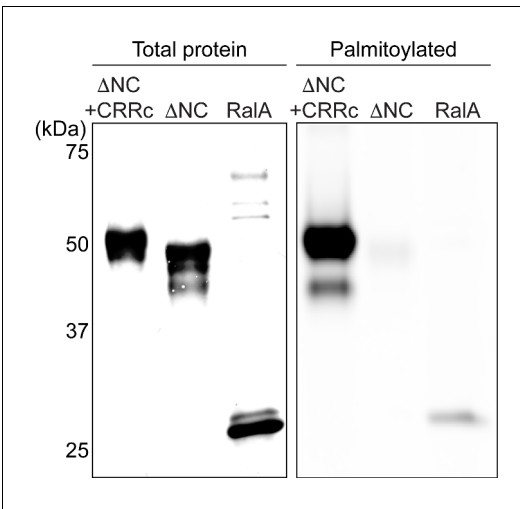

**Figure 9.** CRR of pdP2X7 is palmitoylated in HEK293 cells. HEK293 cells transfected with FLAG-pdP2X7-ΔNC +CRRc, FLAG-pdP2X7-ΔNC, or FLAG-RalA were metabolically labeled with 17-ODYA and pulled down using anti-FLAG antibody. The amount of loaded protein (total protein) was detected by western blotting using anti-FLAG antibody and 17-ODYA incorporation was detected with Alexa Fluor 647-azide (Palmitoylated).
DOI: https://doi.org/10.7554/eLife.31186.015

## Discussion

Permeation of large molecules is a peculiar characteristic of the P2X7 receptor subtype. While numerous studies support that the P2X7 specific CTD is essential for the dye-permeable pore, the role of CTD remains poorly understood. What has also been controversial is whether the P2X7 receptor itself constitutes a dye-permeable pore or indirectly opens other large-conductance channels like pannexin1. In this study, we investigated the channel activity of purified pdP2X7 reconstituted into liposomes in the absence of other cellular components. In contrast to previous studies, we demonstrated that pdP2X7 forms a pore independent of either the CTD or the NTD. This result also provides strong evidence that the P2X7 receptor on its own constitutes a dye-permeable pore.

The reason why previous studies did not detect dye uptake from a C-terminally truncated P2X7 receptor is likely because those studies were based on cell-based systems, which normally include high-levels of cholesterol. Under those conditions, channel activity of a C-terminally truncated P2X7 receptor is strongly attenuated by cholesterol. Indeed, our reconstitution study using mimetic components of the HEK293 cell lipids recapitulated that a C-terminally truncated P2X7 receptor fails to form a dye-permeable pore in the presence of a normal amount of cholesterol (i.e. ~25%); however, removal of cholesterol from those lipids fully recovered the channel activity. We also showed that a C-terminally truncated P2X7 receptor normally functions in cholesterol-deficient Sf9 cells, which, to the best of our knowledge, have not been used for studying P2X7 receptor function. These results highlight that the P2X7 receptor does not require the CTD for its channel activity itself, but it does depend on the CTD for counteracting the inhibitory effect of cholesterol.

Our study is not the first to show an inhibitory role of cholesterol on the P2X7 receptor. Robinson et al demonstrated that MCD treatment substantially enhances hP2X7 mediated dye uptake and that both N- and C-terminal residues are required for such facilitation (*Robinson et al., 2014*). In particular, they found that CRR facilitates P2X7 mediated dye uptake, which is consistent with our results. However, it was unclear whether cholesterol directly binds to the P2X7 receptor or indirectly affects its channel activity. It was also unclear whether the transmembrane domain of the P2X7 receptor plays any role for cholesterol-dependent inhibition. Our *in vitro* binding assay revealed that cholesterol directly interacts with a P2X7 receptor that lacks both N- and C-terminus, indicating that cholesterol likely inhibits channel activity by binding to the transmembrane helices. We also demonstrated that a cholesterol-dependent increase in membrane rigidity is not the primary mechanism of inhibition, as a P2X7 facilitating lipid SM also increases membrane rigidity.

Nevertheless, membrane rigidity does go up as the concentration of cholesterol increases, which makes it difficult to inconspicuously separate these two mechanisms. It would be helpful to identify the cholesterol binding residues for better understanding how membrane rigidity would affect P2X7 receptor function.

Dye uptake activity in the absence of other channels strongly support that the P2X7 receptor itself constitutes a dye-permeable pore. Also, the immediate and monophasic dye uptake supports that the pore of the P2X7 receptor does not dilate. This is consistent with the recent studies demonstrating that NMDG-mediated currents are readily recorded from P2X7 receptor expressing cells without prolonged or repeated application of ATP (*Harkat et al., 2017*). Interestingly, those studies demonstrated that other P2X receptor subtypes including P2X2-4 also give rise to NMDG-mediated currents (*Li et al., 2015*; *Harkat et al., 2017*). These results suggest that the ability to open a dye-permeable pore may actually be a common characteristic of P2X receptors. This is consistent with our current study demonstrating that the P2X7 specific CTD is not required for opening a dye-permeable pore. Furthermore, this idea is also supported by the currently available crystal structures, which present similar overall architectures of the ATP-binding extracellular region and the transmembrane channel for the P2X3, 4 and 7 subtypes (*Kawate et al., 2009*; *Hattori and Gouaux, 2012*; *Karasawa and Kawate, 2016*; *Mansoor et al., 2016*). Notably, permeability of a large molecule such as NMDG seems much less than that of a small ion like Na$^+$, as suggested by Harkat et al. (*Harkat et al., 2017*). This ratio may vary among the different P2X subtypes and the permeability of a large molecule through the P2X7 receptor may be the highest, which could be a reason why the dye-permeable pore has been commonly observed for the P2X7 subtype. Unfortunately, it is technically challenging to compare the permeability of Ca$^{2+}$ and YO-PRO-1 using our reconstitution system because the resulting fluorescent counts depend on many factors such as quantum yields of fluorophores, fluorescence efficiency, and binding properties between the dyes and their substrates.

We also demonstrated that extending the C-terminus of pdP2X7-ΔNC by just nine residues almost completely recovers the dye-uptake activity in the presence of MCD. Because this construct (pdP2X7-ΔNC+CRRc) was efficiently labeled with a palmitic acid analogue in HEK293 cells, it is likely that the facilitating function of CRR derives from palmitoylation. Indeed, mutation of the first two cysteine residues in CRR (C362 and C363) rendered the full-length pdP2X7 incapable of dye-uptake, even in the presence of the entire CTD. What is the potential mechanism for palmitoylated cysteine residues in P2X7 channel facilitation? Given that C362 and C363 are located near the end of the second transmembrane helix (TM2), we speculate that acylation of these two cysteines could alter the tilting angles of TM2. Alternatively, palmitoylated cysteines may serve as an intracellular anchor, which may facilitate the movement required for P2X7 channel opening. While these types of regulation have been suggested for ion channels in general (*Nyholm et al., 2007*; *Shipston, 2014*), the effect of palmitoylation may be particularly strong for P2X receptors, whose membrane pore is one of the simplest with only three pore-lining helices. It is also possible that attached fatty acids may interact with cholesterol bound to the transmembrane helices of the P2X7 receptor, which may reduce the inhibitory effect of cholesterol. A similar mechanism has been proposed for the μ-opioid receptor, whose homodimer may be stabilized through direct interaction between cholesterol and palmitate attached to a nearby cysteine (*Zheng et al., 2012*). Though it would be informative to use a reconstitution system for probing how palmitoylated cysteines control P2X7 channel activity, isolation of a palmitoylated P2X7 receptor is technically challenging because such a protein is prone to aggregation in detergents.

Our reconstitution system allowed us to directly investigate how lipid composition affects P2X7 activity. By systematically testing different types of lipids, we discovered that P2X7 channel opening is facilitated by POPG or SM but not by other common lipids in plasma membrane such as POPC or POPS. Though POPG is a minor component (typically less than 2% of total lipids) for most neuronal and blood cells (*Gottfried, 1967*; *Cornwell et al., 1968*; *Calderon et al., 1995*), this lipid exists at relatively high concentration (~9%) in alveolar macrophage (*Sahu and Lynn, 1977*). Considering that macrophages only contain ~8% cholesterol in the membrane (*Sahu and Lynn, 1977*), the inhibitory action by cholesterol is probably weaker in these cells. This is consistent with a number of previous studies reporting the formation of dye-permeable pores in macrophages (*Steinberg et al., 1987*; *Hickman et al., 1994*; *Falzoni et al., 1995*; *Rassendren et al., 1997*; *Solle et al., 2001*). Also, POPG has been shown to be up-regulated in lymphoma cells that overexpress the v-myc avian myelocytomatosis viral oncogene homolog (*Eberlin et al., 2014*). While it is uncertain whether up-

regulation of POPG is a common phenomenon for other cancer cells, it provides another potential mechanism for P2X7 receptor hyper-activation, which has been implicated in different types of cancer (*Di Virgilio et al., 2009*; *Roger and Pelegrin, 2011*; *Adinolfi et al., 2012*). In contrast to POPG, SM is a major lipid in the plasma membrane; however, SM content highly varies from cell to cell. For example, red blood cells contain 20–30% SM (*Cornwell et al., 1968*) but lymphoblasts contain only 4% (*Gottfried, 1967*). Interestingly, activation of P2X7 receptors have been demonstrated to activate acid sphingomyelinase in microglia, which is responsible for degrading SM in the plasma membrane and triggering IL-1β release (*Bianco et al., 2009*). This negative feedback seems to be an excellent mechanism to control P2X7 receptor activity. Notably, deficiency in acid sphingomyelinase leads to the devastating Niemann-Pick disease that is associated with severe cognitive decline and impaired neuromotor coordination (*Ledesma et al., 2011*; *Rigante et al., 2017*). Though the exact mechanism remains poorly understood, it is possible that accumulation of SM due to the lack of acid sphingomyelinase activity may hyper-activate P2X7 receptors.

We should point out that the presented reconstitution experiments are limited to the biochemically stable P2X7-ΔNC. Though reconstitution of WT P2X7 would be informative, it was technically challenging because this protein was unstable and purified mostly as aggregates. Nevertheless, the previous studies by the Murrell-Lagnado group (*Robinson et al., 2014*) and our cell-based studies using MCD demonstrate that cholesterol also diminishes channel activity of the full-length P2X7. These experiments indicate that the inhibitory effect of cholesterol is not limited to P2X7-ΔNC but also present for the WT P2X7.

In conclusion, we have demonstrated that the P2X7 receptor harbors an intrinsic dye-permeable pore whose activity depends substantially on lipid composition. Because transmembrane helices of the P2X7 receptor are sufficient for the formation of the dye-permeable pore, it is unlikely that the P2X7 specific CTD constitutes a pore or recruits other pore-forming proteins. The CTD instead facilitates its channel activity through palmitoylated cysteines near the end of the pore-lining helix, which contributes to counteract the inhibitory effect of cholesterol. These novel insights imply that activation of P2X7 receptors considerably varies depending on the host cells, which may be a plausible explanation for inconsistent results from previous functional studies.

## Materials and methods

### Construction of expression plasmids

Deletion constructs of P2X7 from giant panda (*Ailuropoda melanoleuca*; pdP2X7; XP_002913164.1) were subcloned into pIE2, pIM2 (*Karasawa and Kawate, 2016*) or pIE5 vector that harbors a FLAG tag at the N-terminus. Point mutations were introduced into constructs via QuikChange site-directed mutagenesis. pdP2X7-ΔNC was subcloned into pNGFP-FB3 vector for expression and purification from Sf9 cells (*Karasawa and Kawate, 2016*).

### Cell line generation

HEK293 (CRL-1573) cell lines were purchased from the American Type Culture Collection (ATCC, Manassas, VA), and therefore were not further authenticated. The mycoplasma contamination test was confirmed to be negative at ATCC.

### Electrophysiology

Patch clamp recording of HEK293 cells expressing pdP2X7 constructs was performed as previously described (*Karasawa and Kawate, 2016*). Briefly, HEK293 cells were transfected with 1 μg of the pdP2X7 constructs using FuGENE6 (Promega, Madison, WI). Cells were used 18–32 hr after transfection for measuring the pdP2X7 receptor activities using the whole cell patch clamp configuration. Membrane voltage was clamped at −60 mV with an Axopatch 200B amplifier (Molecular Devices, Sunnyvale, CA), currents were filtered at 2 kHz (eight-pole Bessel; model 900BT; Devices, Ottawa, IL), and the recording data were sampled at 10 kHz using a Digidata 1440A and pCLAMP 10.5 software (Molecular Devices, Sunnyvale, CA). The extracellular solution contained 147 mM NaCl, 10 mM HEPES, 13 mM Glucose, 2 mM KCl, 0.1 mM CaCl$_2$, (pH 7.3). The pipette solution contained 147 mM NaCl, 10 mM HEPES, 10 mM EGTA, which was adjusted to pH7.0 using NaOH. Extracellular

solutions were rapidly exchanged to the solutions containing 1 mM ATP using a computer-controlled perfusion system (RSC-200; Bio-Logic, France).

## Cellular YO-PRO-1 and Ca$^{2+}$ uptake assay

YO-PRO-1 uptake of HEK293 cells expressing pdP2X7 constructs were measured as previously described (*Karasawa and Kawate, 2016*) with minor modification. pdP2X7 expressing HEK293 cells were plated on poly-D-lysine coated black-walled 96 well plates (Corning, Corning, NY), washed with assay buffer (147 mM NaCl, 2 mM KCl, 0.1 mM CaCl$_2$, 13 mM Glucose, 10 mM HEPES (pH 7.3)), and incubated with or without 5 mM methyl-β-cyclodextran (MCD; Sigma-Aldrich, St. Louis, MO) at 37°C for 1 hr. Cells were washed again and incubated with 5 μM YO-PRO-1 iodide (Thermo Fisher Scientific, Waltham, MA) in assay buffer at 37°C for 10 min. Upon application of 1 mM ATP (final), uptake of YO-PRO-1 was recorded at 37°C with 1 min intervals by following the fluorescence change using Synergy HT multi-detection microplate reader (Bio-Tek, Winooski, VT; Ex: 485 nm/20, Em: 528 nm/20, sensitivity 60). The initial rates of YO-PRO-1 uptake were compared using Origin software (Originlab, Northampton, MA). For YO-PRO-1 uptake of Sf9 cells, 30 μl of P1 virus was used to inoculate 20 ml Sf9 cell suspension at 1.0 × 10$^6$ cells/ml in ESF921 media (Expression systems, Davis, CA). Cells were cultured at 27°C for 24 hr then at 18°C for 24 more hours before harvesting. For each experiment, 200 μl of infected cells were transferred into a well of poly-D-lysine coated black-walled 96 well plates and incubated at 18°C for 24 hr. Cells were washed with assay buffer and incubated with 5 mM MCD in assay buffer at 27°C for 30 min. After washing the cells again with assay buffer, YO-PRO-1 uptake was measured at 25°C as described above. For Ca$^{2+}$ uptake assay, HEK293 cells were incubated with 5 μM Fluo-4 AM in assay buffer at room temperature for 30 min. The cells were washed twice with assay buffer to remove free Fluo-4 AM. After replacing the extracellular solution with assay buffer containing 3 mM CaCl$_2$, Ca$^{2+}$ uptake upon application of 1 mM ATP was recorded at 25°C with 10 s intervals by following fluorescence change as described above.

## Cell surface protein biotinylation

HEK293 cells were plated in six well plates and transfected with 2 μg/well of pdP2X7 constructs in pIE5 vector. Two days after transfection, cells were rinsed twice with ice-cold PBS and incubated with freshly prepared 0.2 mg/ml EZ-Link sulfo-NHS-LC-biotin (Thermo Fisher Scientific) in biotinylation buffer (140 mM NaCl, 5 mM KCl, 2 mM CaCl$_2$, 1 mM MgCl$_2$, 10 mM D-glucose, 10 mM Hepes, pH 7.3) for 1 hr at 4°C. Biotinylation was stopped with 5 mM ammonium chloride and the cells were washed three times with T buffer (25 mM Tris-HCl, 150 mM NaCl, 10 mM EDTA, pH 7.5). Cells were lysed in 350 μl of solubilization buffer (1% TritonX-100, Halt Protease Inhibitor Cocktail (Thermo Fisher Scientific) in T buffer) with agitation for 1 hr at 4°C. Soluble fraction was collected by centrifugation at 21,100 x g for 15 min at 4°C and the biotinylated protein was bound to 30 μl of Strep-Tactin resin (GE Healthcare, Marlborough, MA) with rotation at 4°C for 2 hr. Resin was washed four times with T buffer supplemented with 1% Triton-X100, and the biotinylated protein was eluted with 50 μl of elution buffer (100 mM Tris-HCl (pH 8.0), 150 mM NaCl, 1 mM EDTA, 2.5 mM desthiobiotin, and 1% Triton-X100). The cell lysate prior to Strep-Tactin affinity purification (total) and the pulled down proteins (biotinylated) were applied to 12% SDS-PAGE, transferred onto a nitrocellulose membrane, detected with monoclonal anti-FLAG antibody (1:1,000; Sigma-Aldrich), labeled with an anti-mouse AP-conjugate secondary antibody (1:1,000; Bio-Rad Laboratories, Hercules, CA), and developed using colorimetric AP substrate (Bio-Rad Laboratories).

## Purification of pdP2X7-ΔNC

pdP2X7-ΔNC was expressed in Sf9 insect cells and purified as previously described (*Karasawa and Kawate, 2016*). Briefly, Sf9 cells expressing pdP2X7-ΔNC-EGFP were harvested by centrifugation at 2040 x g, washed with PBS, and resuspended in PBS containing protease inhibitors. The cells were broken by nitrogen cavitation at 600 psi using a 4635 cell disruption vessel (Parr Instrument, Moline, IL). After removing unbroken cells and debris by centrifugation at 12,000 x g for 10 min, the membrane fraction was collected by centrifugation at 185,000 x g for one hour and solubilized in S buffer (2% Triton X-100 (Anatrace, Maumee, OH) in PBS (pH 7.4)) for one hour. The soluble fraction was collected after centrifugation at 185,000 x g for one hour and incubated with Strep-Tactin resin for

30 min using a batch procedure. The resin was transferred into a gravity column and washed with 10 column volumes of W buffer (100 mM Tris-HCl; pH 8.0, 150 mM NaCl, 1 mM EDTA, and 0.5 mM n-Dodecyl β-D-maltoside (DDM)), and pdP2X7-ΔNC-EGFP was eluted with E buffer (100 mM Tris-HCl; pH8.0, 150 mM NaCl, 1 mM EDTA, 2.5 mM desthiobiotin,15% glycerol, and 0.5 mM DDM). The N-terminal EGFP and the strep tag was removed by incubating with human thrombin (1/30 w/w; Haematologic Technologies, Essex Junction, VT) overnight. pdP2X7-ΔNC was isolated by size exclusion chromatography using Superdex 200 (GE Healthcare) in SEC buffer (50 mM Tris-HCl; pH 7.4, 150 mM NaCl, 15% Glycerol, and 0.5 mM DDM). The peak fractions were pooled, and concentrated to 2 mg/ml by Amicon Ultra-4 100K (Merk Millipore, Billerica, MA) and used for reconstitution studies.

## Reconstitution of P2X7 into liposomes

Purified pdP2X7-ΔNC was reconstituted into liposomes composed of a different ratio of phospholipids and/or cholesterol as described previously (*Geertsma et al., 2008*). Briefly, liposomes composed of desired lipid composition was made by mixing with each lipid from 25 mg/ml stock solution in chloroform (Avanti Polar Lipids, Inc., Alabaster, AL) and dried by $N_2$ flow in a glass tube. Dried lipids were dissolved in pentane and dried again under $N_2$ gas. The lipids were resuspended in reconstitution buffer (50 mM Tris-HCl, 150 mM NaCl, 0.1 mM EGTA, pH 7.4) to a concentration of 10 mg/ml by sonication for 10 min. Liposomes were frozen in liquid nitrogen and stored at −80℃. Frozen liposomes (10 mg) were thawed at room temperature, extruded 12 times using a polycarbonate filter (400 nm; Whatman Nucleopore Track-Etched Membranes, Avanti Polar Lipids), and incubated with 6.5 mg of DDM in 1.5 ml reconstitution buffer for 15 min at room temperature. Liposomes and purified pdP2X7-ΔNC were mixed in a 100:1 ratio (w/w) and incubated for 30 min with gentle agitation at room temperature. To remove the detergent, total 400 mg of bio-beads SM2 (Bio-Rad Laboratories) was added to the mixture in four steps (100 mg/step); at each step, the mixture was incubated for 30 min at room temperature, overnight at 4℃, 1 hr at room temperature, and another 1 hr at room temperature, respectively. Bio-beads were removed by passing through a poly-prep chromatography column (Bio-Rad Laboratories). Proteoliposomes were collected by centrifugation at 265,000 x g for 20 min at 4℃, resuspended in reconstitution buffer to a concentration of 10 mg/ml, frozen in liquid nitrogen, and stored in −80℃.

## YO-PRO-1uptake assay using proteoliposomes

A complementary 20-mer DNA pair (40 µM each; 5'-GGATCCCCTGCGTGCTGCTC-3' and 5'-GAG-CAGCACGCAGGGGATCC-3') was incorporated into 10 mg/ml of pdP2X7-ΔNC-reconstituted proteoliposomes by three cycles of freezing in liquid $N_2$ and thawing at room temperature. After extrusion of the proteoliposomes through a polycarbonate filter (200 nm pore size), DNA bound to the outside of liposomes was digested by incubating with 0.2 mg/ml of DNase I (Sigma-Aldrich) and 5 mM $MgCl_2$ for 1 hr at room temperature. The proteoliposomes were washed twice with reconstitution buffer (50 mM Tris, 150 mM NaCl, 0.1 mM EGTA, pH 7.4) by repeating centrifugation (280,000 x g for 20 min at 4℃) and resuspension in reconstitution buffer. For each experiment, 20 µl of proteoliposomes (10 mg/ml) was mixed with 180 µl of YO-PRO-1 in reconstitution buffer (5 µM final concentration) and placed in a well of a 96 well plate. After 10 min incubation at 30℃, 1 mM ATP was added to the well and YO-PRO-1 uptake was recorded by following the fluorescence change at 30℃ with 10 s intervals using a Synergy HT multi-detection microplate reader (Bio-Tek; Ex: 485 nm/20, Em: 528 nm/20, sensitivity 100). Fluorescence intensity was normalized to the maximum fluorescence obtained after solubilizing the liposomes with 1% Triton-X100.

## Ca²⁺ uptake assay using proteoliposomes

Fluo-4 FF (25 µM; Thermo Fisher Scientific) was incorporated into 10 mg/ml of pdP2X7-ΔNC-reconstituted proteoliposomes by three cycles of freezing in liquid $N_2$ and thawing at room temperature. After extrusion of the proteoliposomes through a polycarbonate filter (200 nm pore size), proteoliposomes were separated from free Fluo-4 FF by size exclusion chromatography using 15 ml of Sephadex G-50 resin (GE Healthcare). The proteoliposomes were collected by centrifugation (280,000 x g for 20 min at 4℃) and resuspended with reconstitution buffer to a concentration of 10

mg/ml. Ca$^{2+}$ uptake was recorded as descried in the 'YO-PRO-1 uptake assay using proteoliposomes' section, except that the reconstitution buffer also contained 3 mM CaCl$_2$.

## Membrane rigidity measurement

Membrane rigidity of proteoliposomes was measured as previously described (*Dawaliby et al., 2016*). Briefly, 0.3 mg/ml of pdP2X7-ΔNC-reconstituted proteoliposomes were mixed with 0.2 mg/ml of 1,6-diphenyl-1,3,5-hexatriene (DPH; Sigma-Aldrich) and incubated at 42°C for 30 min for homogeneous incorporation of DPH into the proteoliposomes. Fluorescence anisotropy was measured at 30°C using a FluoroMax-4 fluorimeter (Horiba, Edison, NJ) with excitation and emission wavelengths at 358 nm and 429 nm, respectively. Fluorescence anisotropy <r > was defined as:

$$<r> = \frac{IVV - G*IVH}{IVV + 2*G*IVH}$$

where $I_{VV}$ and $I_{VH}$ are the fluorescence intensities with the excitation polarizer mounted vertically and the emission polarizer mounted vertically or horizontally, respectively. $G$ is defined as:

$$G = \frac{IHV}{IHH}$$

where $I_{HV}$ and $I_{HH}$ are the fluorescence intensities with the excitation polarizer mounted horizontally and the emission polarizer mounted vertically or horizontally, respectively. Molecular models presented in the figures are created using the PyMOL Molecular Graphics System, Version 1.8 Schrödinger, LLC (*Schrodinger, 2015*).

## Cholesterol binding assay

Cholesterol binding assay was performed as previously described with modifications (*Liu et al., 2009*). Purified pdP2X7-ΔNC in SEC buffer (10 µl of 50 µM stock) was mixed with 90 µl of 22-NBD-cholesterol (Thermo Fisher Scientific) in SEC buffer and incubated for 1 hr at 30°C. Methanol was used to dissolve 22-NBD-cholesterol at different concentrations and 3 µl of each was mixed with 87 µl of SEC buffer to obtain final concentrations of 0.1 µM, 0.3 µM, 1 µM, 3 µM, and 10 µM. Fluorescence spectra were recorded at 30°C with an excitation wavelength of 470 nm using a FluoroMax-4 fluorimeter (Horiba). Fluorescence intensities at maximum (527 nm) was plotted against multiple concentrations of 22-NBD-cholesterol.

## Cu(I)-catalyzed azide-alkyne cycloaddition reaction (click chemistry)

Palmitoylation of pdP2X7 constructs were measured using click chemistry as previously described (*Martin, 2013*). Briefly, HEK293 cells in a 100 mm dish were transfected with 10 µg of RalA in pCMV-FLAG vector or pdP2X7 constructs in pIE5 vector using jetPRIME reagent (Polyplus-transfection, France). The culture media was replaced with a fresh DMEM containing 10% FBS one day after transfection. After incubating for another 24 hr, the culture media was replaced with DMEM containing 10% dialyzed-FBS and 100 µM 17-Octadecynoic Acid (17-ODYA; Cayman Chemical, Ann Arbor, MI), and incubated for five hours. Cells were then washed with PBS and lysed with IP lysis buffer (PBS, 1% Triton-X100 and Halt protease inhibitor cocktail (Thermo Fisher Scientific)) for 1 hr at 4°C. After centrifugation at 10,000 x g for 15 min at 4°C, the supernatant was incubated with anti-FLAG M2 beads (Sigma-Aldrich) for 1 hr at 4°C. The beads were washed four times with IP lysis buffer and once with wash buffer (PBS and 0.1% Triton-X100). Washed beads (25 µl) were mixed with 65 µl of wash buffer and 10 µl of freshly premixed click chemistry reagent (final concentrations: 10 µM Alexa Fluor 647-azide (Thermo Fisher Scientific), 1 mM tris(2-carboxyethyl)phosphine (TCEP), 100 µM tris [(1-benzyl-1H-1,2,3-triazol-4-yl)methyl]amine (TBTA), and 1 mM CuSO$_4$) and incubated on a rotator for 1 hr at room temperature in the dark. The beads were washed five times with wash buffer and the bound proteins were eluted at 30°C with 50 µl of elution buffer (PBS, 0.2 mg/ml 3X FLAG Peptide (Sigma-Aldrich), and 0.1% Triton-X100). Eluted protein was treated with sample buffer with 10 mM TCEP prior to SDS-PAGE. Proteins labeled with 17-ODYA were detected by in-gel Alexa fluorescence at 647 nm using Versadoc MP5000 gel imager (Bio-Rad Laboratories). The total protein was detected by western blotting as described above in 'Cell surface protein biotinylation.'

## Acknowledgements

We thank W Greentree, A Nishimura, and M Linder for technical assistance for the palmitoylation assays. We also thank C Westmiller and P Nguyen for discussion and comments on the manuscript.

## Additional information

### Funding

| Funder | Grant reference number | Author |
|---|---|---|
| National Institutes of Health | GM114379 | Toshimitsu Kawate |

The funders had no role in study design, data collection and interpretation, or the decision to submit the work for publication.

### Author contributions

Akira Karasawa, Resources, Data curation, Formal analysis, Validation, Investigation, Methodology, Project administration, Writing—review and editing; Kevin Michalski, Data curation, Validation, Investigation, Methodology, Writing—review and editing; Polina Mikhelzon, Resources, Investigation, Writing—review and editing; Toshimitsu Kawate, Conceptualization, Data curation, Supervision, Funding acquisition, Validation, Visualization, Methodology, Writing—original draft, Project administration, Writing—review and editing

### Author ORCIDs

Toshimitsu Kawate http://orcid.org/0000-0002-5005-2031

### Decision letter and Author response

Decision letter https://doi.org/10.7554/eLife.31186.017
Author response https://doi.org/10.7554/eLife.31186.018

## Additional files

### Supplementary files

• Transparent reporting form
DOI: https://doi.org/10.7554/eLife.31186.016

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
