## [Decision Letter]

Thank you for submitting your article "The P2X7 receptor forms a large pore independent of its intracellular domain but dependent on membrane lipid composition" for consideration by *eLife*. Your article has been favorably evaluated by Richard Aldrich (Senior Editor) and three reviewers, one of whom, Kenton J Swartz (Reviewer #1), is a member of our Board of Reviewing Editors. The following individuals involved in review of your submission have agreed to reveal their identity: Thomas Grutter (Reviewer #2); Mark L Mayer (Reviewer #3).

The reviewers have discussed the reviews with one another and the Reviewing Editor has drafted this decision to help you prepare a revised submission.

Summary:

There has been longstanding confusion about whether the pore of ATP-activated P2X receptor channels can dilate to allow permeation of large cations such as NMDG or fluorophores such as YO-PRO. The idea that P2X receptors progressively dilate in response to continual activation has recently been challenged by demonstrations from multiple groups that the relative permeability of Na to NMDG remains fixed if the concentrations of intracellular ions remains effectively clamped in whole-cell recordings. Whether larger fluorophores can permeate P2X receptor channels remains controversial, with some groups arguing that they permeate through Pannexin hemichannels. This is a remarkable manuscript showing that P2X7 constructs can be purified and reconstituted into liposomes, and that they provide a permeation pathway for small ions such as calcium, but also for large fluorophores such as YO-PRO. With a biochemically well-defined preparation in hand, the authors go on to examine the influence of lipids on P2X7 activity and find that their mainstay deltaNC construct is robustly inhibited by cholesterol. Moreover, they find that the influence of cholesterol on membrane rigidity is unlikely to explain the inhibitory effects, and they demonstrate binding of cholesterol to their P2X7 construct using fluorescence spectroscopy. They then study a C-terminal Cys rich region (CRR) that when re-introduced into their construct partially counteracts the inhibitory influence of cholesterol, and they identify two key Cys residues within the CRR that seem to be most important. Finally, they demonstrate that one or both of these Cys residues are palmitoylated, which may be involved in counteracting the inhibitory influence of cholesterol. Overall this is an exceptionally high quality biochemical study that unequivocally demonstrates that P2X7 provides a permeation pathway for relatively large fluorophores, and that this receptor can be regulated by membrane components. The following are relatively straightforward suggestions for improving an otherwise superb study.

1) One reviewer felt that the jump from presenting the results for expression of WT P2X7 and its deletion constructs in HEK cells in Figure 1 and the preceding text, to the study of proteoliposomes for only the NC deletion construct is awkward. This leaves the reader wondering about the behavior of the WT protein in proteoliposomes. If possible it would be worth presenting data on both Ca and YO-PRO-1 uptake for WT P2X7 reconstituted in liposomes in the second paragraph of the subsection “P2X7-ΔNC alone is sufficient for dye-uptake in proteoliposomes”, as a foundation for the surprising result that the NC deletion construct also shows activity. The rationale for using the deletion construct in most of the experiments is clear, but the modulation of WT channels is also (perhaps more so) important.

2) The following related point was made by another reviewer. The inhibitory influence of cholesterol is largely interrogated in the context of the deltaNC construct. Although the full-length protein may be biochemically poorly behaved, is there any way to demonstrate that cholesterol also inhibits the full-length protein? If not, the authors may want to take a paragraph in the Discussion to flesh this out and at least point out that MCD enhances the activity of the full-length protein in HEK cells (Figure 7), suggesting that the inhibitory influence of cholesterol is also present for the wt protein, albeit less robustly when compared to their deltaNC construct.

3) Have the authors considered testing whether the deltaNC+CRR still binds cholesterol using their fluorescence assay? Seeing a clear shift in apparent affinity would add an additional step in understanding how palmitoylation influence the inhibitory actions of cholesterol.

4) It would be good to have exemplary data for YO-PRO uptake and current density in Figure 1. Also the authors should either select a better trace for the full-length construct because the quality of voltage-clamp is compromised by the voltage errors encountered at such high expression. If all the recordings for the full-length construct have similar issues, it would be good for the authors to point out to the reader that the current densities were so much higher for full-length with their expression construct that it was not possible to obtain well-clamped recordings except for the deletion constructs.

5) The actual composition of different membrane preparations is at times hard to follow. The legend to Figure 3 is clear, but the associated text is not. I could not find a clear statement on how membrane composition was varied in Figure 4.

6) The authors should discuss the large difference in apparent affinity for calcium and YO-PRO uptake. Li et al., 2013 PNAS showed that P2X2 receptors require free ATP for activation, and earlier studies have shown that divalents shift the concentration-dependence for activation of P2X7, consistent with this subtype also requiring free ATP. We suspect that some or all of the difference in the apparent affinity for ATP mediated calcium and YO-PRO uptake shown in Figure 2 is due to the presence or absence of calcium. The methods provide the solution for YO-PRO uptake but not that used for calcium uptake, but the second paragraph of the subsection “P2X7-ΔNC alone is sufficient for dye-uptake in proteoliposomes” implies that 3 mM calcium was present. If free ATP is required for activation of P2X7, the presence of calcium could probably explain some or all of the difference. It would be good to clarify and use the stability constant for calcium binding to ATP to state the predicted shift between the two solutions used in the assays.

Also in the aforementioned paragraph: If 3 mM Ca^2+^ attenuates currents induced by ATP in pdP2X7-∆NC, why does it not block Ca^2+^ entry shown in Figure 2? If this is related to changes in the concentration of free ATP in the presence of calcium then this sentence should be modified.

7) I would recommend replacing 'large pore' throughout the text with something like 'permeation pathway for fluorophores or YO-PRO'. At this point we know the relative permeability of large cations like NMDG is much lower than small ions like sodium, and that the relative permeability of YO-PRO is likely to be much lower still. It’s just that it’s easy to measure accumulation of the fluorophore. Although the permeation mechanism is still mysterious to some of us (and the available structures may be non-native), fluorophore permeation is likely to not involve a pore that is physically large.

8) In the second paragraph of the Introduction it would be good to also cite Pippel et al., 2017 PNAS 114,E2156-65.

9) Figure 2—figure supplement 1: One might have expected random orientation of pdP2X7-∆NC following reconstitution into liposomes; that is 50% outside-out and 50% inside-out. However, the authors provide evidence for only outside-out configuration. How is it possible? Alternatively, it is possible that remaining activity of EndoH, even following heat treatment, can still deglycosylate the receptor. One way to exclude this possibility is to show a control experiment in which heat-treated EndoH is added to samples just before SDS-PAGE running.

10) Subsection “P2X7-ΔNC alone is sufficient for dye-uptake in proteoliposomes”, first paragraph: change to:.… diminished ATP triggered YO-PRO-1 uptake.…

11) In Figure 2 why is there an instantaneous jump in YO-PRO-1 fluorescence, followed by a slower rise? The same trend is seen in other panels, notably supplement Figure 4.

12) Figure 3 legend bold text end of sentence. Change to.… in HEK cell mimetic liposomes.

13) Figure 3 panel D. It would be very useful to draw a horizontal dotted line at the value for the control response, with the individual opaque bars overlying the dotted line, to make it easier for readers to see changes produced by different lipid compositions.

14) For Figure 7 and Figure 8 replace Full by WT (for wild type)?

15) Figure 8 legend: for clarity, given that panel A shows the human sequence, change P2X7 in the text to pdP2X7.

16) Discussion, sixth paragraph, first sentence: Is this true, or are the lipid modulation results applicable to only the NC deletion construct? It would be good to distinguish between lipids and cholesterol.

17) Discussion, last paragraph: Is speculation that the CTD forms the large pore even warranted? Is the amino acid sequence compatible with this forming a membrane spanning, or membrane re-entrant loop?

---

## [Author Response]

1) One reviewer felt that the jump from presenting the results for expression of WT P2X7 and its deletion constructs in HEK cells in Figure 1 and the preceding text, to the study of proteoliposomes for only the NC deletion construct is awkward. This leaves the reader wondering about the behavior of the WT protein in proteoliposomes. If possible it would be worth presenting data on both Ca and YO-PRO-1 uptake for WT P2X7 reconstituted in liposomes in the second paragraph of the subsection “P2X7-ΔNC alone is sufficient for dye-uptake in proteoliposomes”, as a foundation for the surprising result that the NC deletion construct also shows activity. The rationale for using the deletion construct in most of the experiments is clear, but the modulation of WT channels is also (perhaps more so) important.

We agree with the reviewer and have actually attempted many times to reconstitute purified WT channel into liposomes. However, we found it technically challenging to study WT P2X7 in our system, as the full-length protein purifies mostly as aggregates (Author response image 1) and reconstitution efficiency is extremely low. Though the poorly reconstituted WT P2X7 did show weak YO-PRO-1 uptake activity, it was questionable whether the CTD remained structurally intact in these samples. Given that the ΔNC is biochemically stable, we suspect that the poor behavior of WT P2X7 derives from the unstable CTD, which probably promotes protein aggregations through unstructured regions and/or highly acylated cysteine residues.

**Author response image 1. respfig1:** Both the full-length pdP2X7 and pdP2X7-ΔNC+CRR purify mostly as aggregate. SEC traces after thrombin digestion of the full-length (**A**) and ΔNC+CRR (**B**) constructs.

2) The following related point was made by another reviewer. The inhibitory influence of cholesterol is largely interrogated in the context of the deltaNC construct. Although the full-length protein may be biochemically poorly behaved, is there any way to demonstrate that cholesterol also inhibits the full-length protein? If not, the authors may want to take a paragraph in the Discussion to flesh this out and at least point out that MCD enhances the activity of the full-length protein in HEK cells (Figure 7), suggesting that the inhibitory influence of cholesterol is also present for the wt protein, albeit less robustly when compared to their deltaNC construct.

As mentioned above, the poor behavior prevents us from investigating the function of WT P2X7 using our reconstitution system. We have added a new paragraph in Discussion to explain this point. We also included a discussion about the effect of cholesterol on WT P2X7 in the same paragraph.

3) Have the authors considered testing whether the deltaNC+CRR still binds cholesterol using their fluorescence assay? Seeing a clear shift in apparent affinity would add an additional step in understanding how palmitoylation influence the inhibitory actions of cholesterol.

Yes, we have attempted to test the DNC+CRR construct for cholesterol binding. Like WT P2X7, however, the DNC+CRR protein purifies mostly as aggregates (Author response image 1) probably due to palmitoylation at the CRR. While we have tried to remove the attached acyl group with hydroxylamine and DTT, we failed to prevent aggregation. We also tried supplementing the purification buffer with a higher concentration of glycerol or truncating the CRR down to C370 (i.e. CRRc), but failed to obtain stable species.

4) It would be good to have exemplary data for YO-PRO uptake and current density in Figure 1. Also the authors should either select a better trace for the full-length construct because the quality of voltage-clamp is compromised by the voltage errors encountered at such high expression. If all the recordings for the full-length construct have similar issues, it would be good for the authors to point out to the reader that the current densities were so much higher for full-length with their expression construct that it was not possible to obtain well-clamped recordings except for the deletion constructs.

We now include exemplary YO-PRO-1 uptake data in Figure 1 (P2X7-mediated currents are shown in Figure 1—figure supplement 1). We have recorded the full-length P2X7 mediated currents from more than 30 cells, all of which showed compromised currents as represented in the figure. This is most likely due to cell-blebbing, which is one of the well-known characteristics of the P2X7 receptor. As suggested by the reviewer, we have included the explanation in the figure legend.

5) The actual composition of different membrane preparations is at times hard to follow. The legend to Figure 3 is clear, but the associated text is not. I could not find a clear statement on how membrane composition was varied in Figure 4.

Thank you for pointing this out. We have included more explicit description in the Figure 4 legend.

6) The authors should discuss the large difference in apparent affinity for calcium and YO-PRO uptake. Li et al., 2013 PNAS showed that P2X2 receptors require free ATP for activation, and earlier studies have shown that divalents shift the concentration-dependence for activation of P2X7, consistent with this subtype also requiring free ATP. We suspect that some or all of the difference in the apparent affinity for ATP mediated calcium and YO-PRO uptake shown in Figure 2 is due to the presence or absence of calcium. The methods provide the solution for YO-PRO uptake but not that used for calcium uptake, but the second paragraph of the subsection “P2X7-ΔNC alone is sufficient for dye-uptake in proteoliposomes” implies that 3 mM calcium was present. If free ATP is required for activation of P2X7, the presence of calcium could probably explain some or all of the difference. It would be good to clarify and use the stability constant for calcium binding to ATP to state the predicted shift between the two solutions used in the assays.

We indeed included 3 mM CaCl_2_ in the assay buffer, which was necessary to obtain robust fluorescence signals. We have updated the method accordingly. We also calculated the concentrations of free ATP in our experimental condition using Max Chelator (http://maxchelator. stanford.edu/) and put those numbers in Figure 2. As pointed out by the reviewer, the EC50 value based on the fee ATP concentrations (29 μM) appeared to be similar to that in YO-PRO-1 uptake assay (21 μM), supporting that free ATP is required to activate the P2X7 channel. We have included this discussion in the main text.

Also in the aforementioned paragraph: If 3 mM Ca^2+^ attenuates currents induced by ATP in pdP2X7-∆NC, why does it not block Ca^2+^ entry shown in Figure 2? If this is related to changes in the concentration of free ATP in the presence of calcium then this sentence should be modified.

Thank you for pointing this out. As mentioned above, the main reason for the low currents in whole cell recordings are likely due to the diminished amount of free ATP. We restated the indicated statement accordingly.

7) I would recommend replacing 'large pore' throughout the text with something like 'permeation pathway for fluorophores or YO-PRO'. At this point we know the relative permeability of large cations like NMDG is much lower than small ions like sodium, and that the relative permeability of YO-PRO is likely to be much lower still. It’s just that it’s easy to measure accumulation of the fluorophore. Although the permeation mechanism is still mysterious to some of us (and the available structures may be non-native), fluorophore permeation is likely to not involve a pore that is physically large.

We agree with the reviewer and substituted "large pore" with "dye-permeable pore" including the title.

8) In the second paragraph of the Introduction it would be good to also cite Pippel et al., 2017 PNAS 114,E2156-65.

We now cite the suggested paper on page 3. Thanks.

9) Figure 2—figure supplement 1: One might have expected random orientation of pdP2X7-∆NC following reconstitution into liposomes; that is 50% outside-out and 50% inside-out. However, the authors provide evidence for only outside-out configuration. How is it possible? Alternatively, it is possible that remaining activity of EndoH, even following heat treatment, can still deglycosylate the receptor. One way to exclude this possibility is to show a control experiment in which heat-treated EndoH is added to samples just before SDS-PAGE running.

We are confident that the orientation of reconstituted pdP2X7-∆NC is mostly outside out. Inactivation of EndoH using our method seemed to be efficient, as other membrane proteins we have reconstituted in our lab showed 50/50 orientations using the same technique. Perhaps the large extracellular domain and the relatively small transmembrane domain facilitate unidirectional incorporation of pdP2X7-∆NC. We nevertheless performed the suggested experiment, which compellingly verifies the outside out orientation. This new result is now shown in Figure 2—figure supplement 1.

10) Subsection “P2X7-ΔNC alone is sufficient for dye-uptake in proteoliposomes”, first paragraph: change to:.… diminished ATP triggered YO-PRO-1 uptake.…

Done.

11) In Figure 2 why is there an instantaneous jump in YO-PRO-1 fluorescence, followed by a slower rise? The same trend is seen in other panels, notably supplement Figure 4.

The instantaneous jump is due to a combination of the low sampling frequency (one measurement every 10 sec) and plate shaking (2-3 sec) after the addition of ATP. This technical limitation did not interfere with our analysis.

12) Figure 3 legend bold text end of sentence. Change to.… in HEK cell mimetic liposomes.

Not all lanes show P2X7 reconstitution in HEK mimetic liposomes. To avoid potential confusion, we changed it to "…liposomes with different lipid compositions."

13) Figure 3 panel D. It would be very useful to draw a horizontal dotted line at the value for the control response, with the individual opaque bars overlying the dotted line, to make it easier for readers to see changes produced by different lipid compositions.

We agree and updated the figure.

14) For Figure 7 and Figure 8 replace Full by WT (for wild type)?

We agree. We have updated those figures.

15) Figure 8 legend: for clarity, given that panel A shows the human sequence, change P2X7 in the text to pdP2X7.

Done. Thanks.

16) Discussion, sixth paragraph, first sentence: Is this true, or are the lipid modulation results applicable to only the NC deletion construct? It would be good to distinguish between lipids and cholesterol.

Our wording was probably unclear. We restated the sentence that reads "inhibitory action by cholesterol is probably weaker in these cells."

17) Discussion, last paragraph: Is speculation that the CTD forms the large pore even warranted? Is the amino acid sequence compatible with this forming a membrane spanning, or membrane re-entrant loop?

While there is no predicted membrane spanning domain in the CTD, membrane reentrant loop could exist. It is also possible that this domain might assemble into a barrel like pore. Conversion of a soluble domain into a membrane spanning pore has been recognized for several types of toxins.